# T1-PILOT: Physics-Informed Learned Optimized Trajectories for T1 Mapping Acceleration

**Tamir Shor**[1] (iD)                  TAMIR.SHOR@CAMPUS.TECHNION.AC.IL
**Moti Freiman**[1] (iD)                 MOTI.FREIMAN@TECHNION.AC.IL
[1] *Technion*
**Chaim Baskin**[2]                       CHAIMBASKIN@BGU.AC.IL
[2] *Ben-Gurion University*
**Alex Bronstein**[1,3]                    BRON@CS.TECHNION.AC.IL
[3] *Institute of Science and Technology Austria*

**Editors:** Accepted for publication at MIDL 2026

## Abstract

Cardiac T1 mapping provides critical quantitative insights into myocardial tissue composition, enabling the assessment of pathologies such as fibrosis, inflammation, and edema. However, the inherently dynamic nature of the heart imposes strict limits on acquisition times, making high-resolution T1 mapping a persistent challenge. Compressed sensing (CS) approaches have reduced scan durations by undersampling k-space and reconstructing images from partial data, and recent studies show that jointly optimizing the undersampling patterns with the reconstruction network can substantially improve performance. Still, most current T1 mapping pipelines rely on static, hand-crafted masks that do not exploit the full acceleration and accuracy potential. Furthermore, most existing methods do not levarage the physical T1 decay model in optimization. In this work, we introduce T1-PILOT: an end-to-end method that explicitly incorporates the T1 signal relaxation model into the sampling–reconstruction framework to guide the learning of non-Cartesian trajectories, cross-frame alignment, and T1 decay estimation. Through extensive experiments on the CMRxRecon dataset, T1-PILOT significantly outperforms several baseline strategies (including learned single-mask and fixed radial or golden-angle sampling schemes), achieving higher T1 map fidelity at greater acceleration factors. In particular, we observe consistent gains in PSNR and VIF relative to existing methods, along with marked improvements in delineating finer myocardial structures. Our results highlight that optimizing sampling trajectories in tandem with the physical relaxation model leads to both enhanced quantitative accuracy and reduced acquisition times. Code for reproducing all experiments and results is available at https://github.com/tamirshor7/T1-PILOT

**Keywords:** Cardiac T1 Mapping, Trajectory Optimization and Reconstruction, Physics-Informed Deep-Learning.

## 1. Introduction

Cardiac MRI T1 mapping quantifies the T1 relaxation time of cardiac tissues, which provides critical insights into tissue composition and plays a key role in diagnosing various pathologies, such as tumors, fibrosis, and diffuse myocardial inflammation (Burt et al., 2014). One of the most prevalent T1 mapping methods is Modified Look-Locker Inversion Recovery (MOLLI) (Messroghli et al., 2004), where multiple images are acquired at different

inversion times following an inversion pulse, and the signal recovery is fitted to an exponential model to estimate the T1 decay map. Despite its success, MOLLI's dependence on multiple breath-holds and heart rate variations makes it highly sensitive to motion artifacts, limiting the feasibility of long scan times often required for high-resolution MRI (Xue et al., 2012; Hanania et al., 2024).

Compressed sensing (CS) (Lustig et al., 2007) has become a powerful approach for addressing these long scan times and motion sensitivity in MOLLI-based T1 mapping. By leveraging the sparsity of MR images in the frequency domain, CS enables precise image reconstruction from highly undersampled k-space data, significantly accelerating the acquisition while preserving essential clinical information. This results in reduced breath-hold durations and improved temporal resolution, alleviating issues related to prolonged scan times and motion artifacts (Stainsby et al., 2014; Lyu et al., 2025; Paajanen et al., 2023). However, most existing methods depend on fixed, hand-selected (Paajanen et al., 2023; Lyu et al., 2025) acquisition schemes. Some research has explored optimizing undersampling strategies alongside downstream T1 map estimation, but these efforts are predominantly confined to using a single, shared optimization scheme across frames (Chaithya et al., 2022; Li et al., 2012). For instance, (Zhang et al., 2015) leverages low-rank constraints and parallel imaging to optimize pulse-sequence parameters, while (Stainsby et al., 2014) employs PCA-guided compressed-sensing reconstructions to shorten acquisition times. Deeplearning approaches have also been explored for acceleration; (Hanania et al., 2024) tackles retrospective motion-artifact correction, and (Guo et al., 2022) proposes an MLP that estimates T1 decay from as few as four T1-weighted images. Nevertheless, these methods either do not fully integrate compressed-sensing (Lustig et al., 2007; Weiss et al., 2019) or rely on fixed undersampling masks (Lyu et al., 2025; Chaithya et al., 2022; Tran-Gia et al., 2015) – a limitation also noted in (Paajanen et al., 2023).

Recent advancements suggest that learned non-Cartesian per-frame k-space undersampling masks can substantially improve acceleration, especially for spatially or temporally sequential MRI data (Shor et al., 2023; Yiasemis et al., 2024; Shor et al., 2024). Prior work shows that such per-frame optimization generally outperforms pipelines using single-frame or Cartesian-based acquisition (Weiss et al., 2019; Wang et al., 2022; Shor et al., 2023). However, even in approaches that do adopt a learned sampling trajectory for T1 decay estimation (Zhang et al., 2024), the actual decay model itself is often not integrated as a driving constraint in the acquisition–reconstruction optimization. Such methods, that ignore the T1-decay model and directly regress the underlying T1 map, require optimization in higher dimension and weaker regularization for the feasible set of potential solutions, that must be learned implicitly. In this paper, we demonstrate how this choice leads to sub-optimal results.

To address these gaps, in this work we opt to develop a T1 acceleration model that explicitly integrates the T1 signal relaxation model as a constraint in optimization, guiding both the learned acquisition and the subsequent image reconstruction. To do so, we introduce T1-PILOT, a novel pipeline for the retrospective simulation-based joint optimization of T1 mapping estimation and physically feasible non-Cartesian per-frame k-space undersampling trajectories. We demonstrate that this approach enables accurate T1 mapping from highly undersampled data, outperforming both constant and learned sampling schemes that do not fully exploit the relaxation model.

We conducted extensive experiments on the CMRxRecon dataset (Wang et al., 2023), comparing T1-PILOT against multiple baselines, including fixed radial and golden-angle undersampling schemes as well as single learned trajectories. Our results show superior performance in terms of PSNR and VIF, highlighting that explicitly modeling the T1 relaxation signal in the undersampling–reconstruction pipeline yields both enhanced quantitative accuracy and significant reductions in acquisition times.

## 2. Method

### 2.1. Problem Definition

Given a sequence of $N$ T1-weighted images $\mathcal{X} = \{x_{t_i}\}_{i=1}^N \in \mathbb{R}^{N \times H \times W}$ sampled at respective inversion times $T = \{t_i\}_{i=1}^N \in \mathbb{R}^N$, the T1 mapping objective is to fit the exponential T1-decay curve of the sequence, parameterized by $A, B, T_1^* \in \mathbb{R}^{\mathbb{H} \times \mathbb{W}}$, according to the decay model:

$$\tilde{x}_{t_i} = A - B \cdot e^{-\frac{t_i}{T_1^*}} \tag{1}$$

In this work we study MOLLI sequences, where the underlying T1 map can be thereafter received by applying a linear correction $T_1 = T_1^* \cdot (\frac{B}{A} - 1)$(Slavin, 2014).

While parameterizing and directly optimizing the decay parameters $A, B, T_1^*$ is possible with classical least-squares approaches, this choice decouples the optimization task of the three decay components, limiting the utilization of inductive biases and regularization that can be injected by using a neural network. Hence, in this work we follow the currently more-common approach (e.g. (Guo et al., 2022; Zhang et al., 2024)) of a neural-network outputting the decay parameters based on T1-weighted input-sequence conditioning - $\mathcal{M} : \mathbb{R}^{N \times H \times W} \mapsto \mathbb{R}^{3 \times H \times W}$. The rationale behind this choice is that, as shown in section 3, such learned neural mappings allow implicit regularization of the exponential regression task, and can be trained across multiple samples to allow efficient learning of data priors, which may alleviate (or completely nullify) the need for per-sample decay-parameter optimization. Denoting by $\theta$ the set of learnable parameters of the neural decay estimation model $\mathcal{M}$, the optimization task for decay estimation can be formulated according to equation 2:

$$argmin_\theta \sum_{t_i \in T} \|x_{t_i} - (A_\theta - B_\theta \cdot e^{-\frac{t_i}{T_{1\,\theta}^*}})\|_2 \tag{2}$$

To *accelerate* T1 map estimation, we opt to optimize a set of $N$ k-space undersampling masks $K \in \mathbb{R}^{N \times n \times m}$, where $n$ is the number of RF excitation pulses (namely, RF *shots*) and $m$ is the number of k-space sampling points per shot. Denote by $\mathcal{F}_K(x_{t_i})$ the downsampling operator of the $i^{\text{th}}$ frame $x_{t_i} \in \mathbb{R}^{H \times W}$ according to the undersampling mask $K$. Importantly, unlike previous works where acquisition trajectory learning is guided solely by a per-frame reconstruction of the T1-weighted sequence (Zhang et al., 2024; Wang et al., 2022), to best-adapt our acquisition set $K$ for our task at hand, in this work our we design our learning objective so that the optimization of $K$ is explicitly informed by the downstream fit to the underlying physical decay model (eq. 1). Our optimization objective therefore augments the one from eq. 2 with an additional set of parameters, $\psi$, that governs our learned undersampling masks. Our objective for joint acceleration and decay-estimation

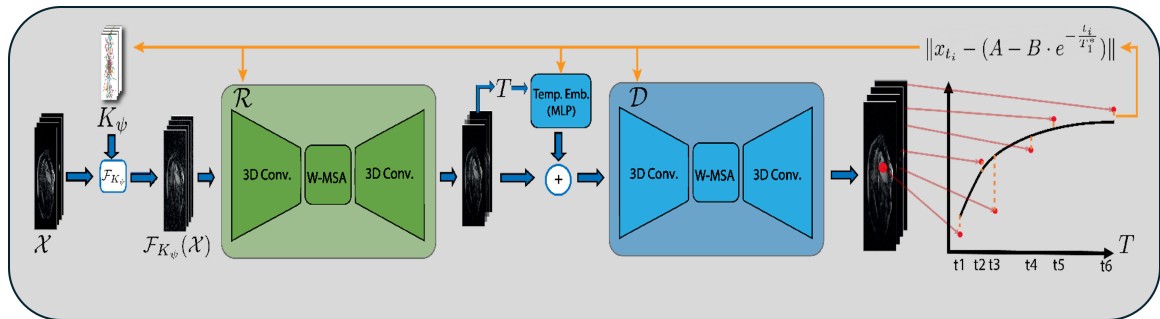

Figure 1: **T1-PILOT Pipeline** - Blue arrows denote the forward pass and orange arrows denote backpropagation. The pipeline is composed of three components - (1) the undersampling module used in training to simulate k-space undersampling with learned trajectories (leftmost), (2) the reconstruction module (green) and (3) the decay estimation module (blue).

can therefore be formulated as:

$$argmin_{\theta,\psi} \sum_{t_i \in T} \|x_{t_i} - (A_{\theta,\psi} - B_{\theta,\psi} \cdot e^{-\frac{t_i}{T_1^*{}_{\theta,\psi}}})\|_2 \tag{3}$$

Where $(A_{\theta,\psi}, B_{\theta,\psi}, T_1^*{}_{\theta,\psi}) = \mathcal{M}_\theta(\mathcal{F}_{K_\psi}(\mathcal{X}))$ and $\mathcal{F}_{K_\psi}(x_{t_i})$ is the downsampling operator of $x_{t_i}$ by the acquisition-set $K$ parameterized by $\psi$.

## 2.2. Optimization

A key consideration in developing a feasible learned k-space acquisition set is that, unlike the decay model parameters governed by $\theta$, $K_\psi$ must be optimized as an acquisition set shared across the entire dataset. This is because, while $\mathcal{M}_\theta$ can be finetuned per input sample $\mathcal{X}$ retrospectively to the scan, the full k-space data would not be available at inference time. The nature of the task at hand may render the joint optimization of $\mathcal{M}_\theta$ and $K_\psi$ sub-optimal, since the decay objective (eq. 3) is highly-related to sample-specific features, and gradients backpropagated from it supply overly noisy feedback to the sampling parameters $\psi$. This challenge is further intensified in non-Cartesian trajectory learning, as the training process is inherently noisier due to the larger number of learned acquisition parameters.

Relying on prior work, where optimization scheduling has successfully addressed these challenges within the context of MRI reconstruction (Shor et al., 2023), we propose a 3-stage optimization schedule, gradually shifting the optimization focus from reconstruction to decay estimation. In the following sections we layout our proposed optimization approach, also illustrated in fig 1.

### 2.2.1. RECONSTRUCTION-GUIDED PRE-TRAINING

- To obtain a well-initialized set of acquisition trajectories, we first train a reconstruction model $\mathcal{R}_\zeta$ dedicated solely to trajectory optimization and reconstruction. Namely, at this

stage our learning objective is formulated as:

$$\min_{\zeta,\psi} \|\mathcal{X} - \mathcal{R}_\zeta(\mathcal{F}_{\mathcal{K}_\psi}(\mathcal{X})\| \tag{4}$$

For $\mathcal{R}_\zeta$ and $K_\psi$ we adopt TEAM-PILOT (Shor et al., 2024), originally proposed for general-purpose joint optimization of k-space acquisition and reconstruction. We choose this model due to its excellent performance on multi-frame MRI reconstruction, incorporation of kinematic scanner constraints and non-Cartesian acquisition parametrization.

TEAM-PILOT parametrizes non-Cartesian sampling trajectories by optimizing a set of 2D control points ($\psi \in \mathbb{R}^{n \times n \times m \times 2}$ in eq. 3 notations), through which the resulting k-space acquisition trajectory is interpolated via a differentiable SPLINE. Machine-related kinematic constraints are imposed according to (Chauffert et al., 2016). To efficiently process 3D MRI data, the reconstruction model is composed of blocks of 3D-convolutional layers interleaved with W-MSA (Liu et al., 2022) attention layers.

### 2.2.2. DECAY OPTIMIZATION

After pre-training $\mathcal{R}_\zeta$, we initialize another TEAM-PILOT block $\mathcal{D}_\xi$, acting as the decay-parameter estimation model. To condition our decay model over different sets of inversion times $T$ (varying across different sequences), we extend $D_\xi$ with a small MLP to learn temporal embeddings added to each reconstructed input sample $\mathcal{R}(\mathcal{F}_{K_\psi}(x_{t_i}))$. We optimize both the decay and reconstruction model according to eq. 3, where $\mathcal{M}_{\theta=\{\zeta,\xi\}}(\mathcal{X}) = \mathcal{D}_\xi(\mathcal{R}_\zeta(\mathcal{F}_{\mathcal{K}_\psi}(\mathcal{X})), T)$.

### 2.2.3. PER-SAMPLE RECONSTRUCTION & DECAY REFINEMENT

T1 mapping is, at core, an exponential regression task. Unlike other common tasks, such as reconstruction, this fundamental trait enables the usage of the exponential fitting error as a strong regularization signal for further optimization at inference time. While the global optimization strategies described above effectively capture population-level priors and generalizable acquisition patterns, they may lack the specificity required to resolve unique, high-frequency anatomical details or subtle pathological variations inherent to individual subjects. A static, globally-shared model may therefore struggle to maximally exploit the consistency required by the strict T1 relaxation constraints.

In both reconstruction and decay stages of T1-PILOT, all parameters are optimized across sequences from the entire dataset. This allows for the optimization of a decay model and acquisition set that generalize across all samples, as well as for the leveraging of global data-shared priors that can help expedite convergence. As we demonstrate in section 3, these two optimization stages suffice for accurate and substantially accelerated T1-Mapping. Nonetheless, since the decay model is sample-specific, we also propose an optional third stage of decay refinement. At this stage, the acquisition set $K$ remains frozen, whilst the decay-and reconstruction parameters $\theta = \{\zeta, \xi\}$ are finetuned (according to eq. 3) for a specific input sequence of $N$ T1-weighted images.

## 3. Results

### 3.1. Data

In all experiments, we make use of the CMRxRecon dataset (Wang et al., 2023) dataset, containing raw, multi-slice and multi-contrast fully-sampled k-space collected from 300 patients. For each patient and slice, 9 T1-weighted images have been acquired with a 3T MRI scanner. Full acquisition protocols are in (Wang et al.). The pre-trained TEAM-PILOT models mentioned in section 3.2 have been trained on the augmented version of the OCMR dataset (Chen et al., 2020) published in (Shor et al., 2023).

### 3.2. Experimental Setup

Our work primarily aims to demonstrate that combining non-Cartesian k-space acquisition learning with a decay objective-guided optimization process enables significantly improved T1 mapping acceleration. This approach outperforms those in previous studies, which either do not learn the k-space acquisition scheme (e.g. those surveyed in (Lyu et al., 2025)), use a single acquisition scheme (learned or fixed) shared across all frames (e.g., (Paajanen et al., 2023)), or apply multi-frame learning without directly integrating the decay objective into the acquisition optimization (e.g., (Zhang et al., 2024)). To achieve this, we compare our method with 4 representative baseline optimization approaches -

1. *Radial* - The k-space acquisition set is a single radial acquisition mask shared across all frames.

2. *GAR* - As a fixed non-Cartesian per-frame acquisition scheme we use the Golden-Angle Ratio (GAR) 3D acquisition scheme proposed by (Zhou et al., 2017). This method proposes an array angularly-shifted per-frame radial acquisition masks.

3. *Single* - We learn a single, non-Cartesian acquisition mask shared across all frames, similarly to (Weiss et al., 2019; Wang et al., 2022).

4. *Reconstruction* - We learn per-frame non-Cartesian acquisition masks (similarly to our proposed method), however include only the reconstruction objective throughout the entire optimization process from section 2.2.

For a fair comparison, we train our method and all four baselines using the modeling and optimization schedule detailed in section 2.2. Each baseline is trained with 16,32 and 64 shots (RF excitation pulses) per-frame, where each pulse is represented by 513 k-space sampling points. We resize every image to spatial dimensions $144 \times 384$, making the acceleration factor in every $n$-shot experiment $\frac{144 \cdot 384}{n \cdot 513}$. Since ground-truth T1 Maps are not available, to measure mapping performance in each experiment we train a separate decay-estimation model for 150 epochs on the entire fully-sampled data, and treat this model's estimations as ground-truth maps. For this model, we only train $\mathcal{D}_\zeta$ with the decay objective from eq. 3, without the added reconstruction logic.

Following (Shor et al., 2024), for each experiment we train for 150 epochs of reconstruction pre-training, 200 epochs of decay optimization and 3000 iterations of per-sample finetuning, on a single NVIDIA-A6000 GPU. To expedite training and attain better reconstruction

priors, in the reconstruction stage, we initialize $\mathcal{R}$ with a TEAM-PILOT model pre-trained on the larger augmented-OCMR dataset from (Shor et al., 2023), for another 150 epochs separately for each experiment. For similar reasons, in the decay optimization stage, we initialize the decay model $\mathcal{M}$ as the decay-estimation model trained on fully-sampled data. We use the CMRxRecons test-set for testing, and its training set is internally split to training and validation by a 0.8 ratio.

In our training setting, the base reconstruction and decay-estimation networks, without learned acquisition, contain approximately 1.6M trainable parameters. Incorporating learned acquisition adds parameters only through the trajectory representation - each learned shot introduces $N \times m \times 2$ parameters, where $N$ is the number of frames, $m$ is the number of control points per shot, and the factor of 2 accounts for the two spatial coordinates. In our setting ($N = 9, m = 513$), this corresponds to roughly $9.2k$ parameters per shot. Consequently, the learned-acquisition variants add between approximately 145k parameters (for 16 shots) and 590k parameters (for 64 shots), which is small relative to the base network size.

### 3.3. Quantitative Results

Table 1 depicts the quality of optimized decay-model estimation across varying acceleration factors, for our method and considered baselines. Following (Shor et al., 2024), we report mean PSNR and VIF(Sheikh and Bovik, 2006) in all experiments. We report performance according to those metrics in two layouts — First, under sub-header *T1-Map*, quality metrics are measured between T1 maps estimated by each baseline and the maps attained from the model trained on fully-sampled data. Results in this setting reflect the acceleration efficacy of each method by comparing the potential of approximating the performance of a model that utilizes fully-sampled data. This metric also poses a more-direct estimation of the quality of downstream map estimation, in the absence of ground-truth data. Second, to estimate our results in a model-independent manner, under *Decay* we also report PSNR and VIF between the original sequence $x_{i_t}$ and decay-model output sequence $\tilde{x_{i_t}}$ (eq. 1). We include results with and without per-sample finetuning (stage 3 in section 2.2) to assess the performance advantage attained in per-sequence optimization. Since the finetuning stage is performed based on the decay objective, we do not report post-finetuning results for the reconstruction-only baseline. While per-sample reconstruction finetuning for this baseline has been considered, our findings indicate that such sample-specific optimization for reconstruction severely harms decay-estimation performance.

In our setting, nominal acquisition time is proportional to the number of RF shots per frame and the number of frames. Since all methods compared at a given shot setting (16, 32, or 64 shots) use the same number of frames and identical hardware constraints, they therefore have identical nominal acquisition time within each column of Table 1. Differences in performance at a fixed shot count thus reflect acquisition efficiency rather than longer scan duration.

Table 1 exhibits two major trends - First, our method exceeds all baselines in acceleration performance, according to both considered quality metrics and comparison layouts. Specifically, our method's performance in map reconstruction *without* the utilization of per-sample finetuning is comparable or superior to that of other methods *with* the additional

Table 1: **Decay-estimation results comparison**.

| Method | Finetuning | 16 Shots | | | | 32 Shots | | | | 64 Shots | | | |
|---|---|---|---|---|---|---|---|---|---|---|---|---|---|
| | | Decay | | T1-Map | | Decay | | T1-Map | | Decay | | T1-Map | |
| | | PSNR | VIF | PSNR | VIF | PSNR | VIF | PSNR | VIF | PSNR | VIF | PSNR | VIF |
| Radial | ✗ | 29.95 | 0.512 | 18.65 | 0.168 | 32.28 | 0.664 | 22.93 | 0.284 | 34.36 | 0.739 | 24.54 | 0.391 |
| | ✓ | 38.33 | 0.873 | 21.1 | 0.543 | 39.02 | 0.902 | 26.98 | 0.706 | 39.04 | 0.903 | 27.28 | 0.658 |
| GAR | ✗ | 33.87 | 0.711 | 23.54 | 0.3 | 35.47 | 0.786 | 23.95 | 0.421 | 36.35 | 0.835 | 24.43 | 0.478 |
| | ✓ | 38.84 | 0.893 | 25.76 | 0.627 | 38.71 | 0.895 | 27.62 | 0.66 | 38.97 | 0.9 | 27.89 | 0.69 |
| Single | ✗ | 33.33 | 0.695 | 23.98 | 0.286 | 34.71 | 0.758 | 24.57 | 0.367 | 35.76 | 0.801 | 25.1 | 0.458 |
| | ✓ | 38.82 | 0.895 | 26.24 | 0.641 | 39.03 | 0.901 | 27.83 | 0.684 | 39.04 | 0.904 | 27.83 | 0.682 |
| Recon. | ✗ | 35.57 | 0.79 | 24.8 | 0.433 | 36.22 | 0.81 | 25.4 | 0.476 | 37.03 | 0.853 | 25.93 | 0.513 |
| T1-PILOT (Ours) | ✗ | 35.74 | 0.792 | 25.38 | 0.418 | 36.45 | 0.819 | 26.41 | 0.57 | 37.19 | 0.854 | 26.52 | 0.581 |
| | ✓ | **39.04** | **0.902** | **28.62** | **0.715** | **39.1** | **0.905** | **28.81** | **0.723** | **39.22** | **0.907** | **28.84** | **0.726** |

per-sample optimization. Our method offers a boost of at around 2 dB in PSNR against all baselines when finetuning is not incorporated, and around 2.5 dB when finetuning is used. Notably, our method proves especially favorable according to the map comparison layout (under *T1-Map* in the table), that is more-directly correlated with the acceleration potential and downstream objective. We attribute this result to our more-potent parametrization of the acquisition set $K$, as well as to the structured incorporation of decay and reconstruction objectives along training. Importantly, this also implies that our method achieves similar or better T1 accuracy at 16 shots compared to baselines at 32 or even 64 shots, corresponding to approximately 2×–4× reduction in acquisition time for comparable quantitative performance.

The second trend we observe is that the two approaches reliant on multi-frame acquisition-learning (namely, our method and the Reconstruction-only baseline) exhibit substantially favorable performance when per-sample finetuning is not applied. We relate this generalizability potential to efficient learning of data-priors during training.

### 3.4. Qualitative Results

Figure 2 illustrates T1 maps estimated by our method and by each of the four baselines trained for acceleration factors of 64 shots per-frame, compared to the map derived from fully-sampled data (upper-left). Maps are shown both in their entirety and with focus on the myocardium. Our method yields sharper and more accurate estimations, surpassing baselines in capturing fine details and subtle variations, particularly in smaller regions of the map (framed in red). We attribute this improvement to the effectiveness of learned, multi-frame non-Cartesian undersampling, which prioritizes high-frequency components essential for preserving such subtle features. As evident from the figure, baseline sampling patterns with non-learned (GAR, Radial) or single-frame (denoted *Single*, bottom-right) acquisition patterns struggle in identifying such patterns.

### 3.5. Learned Acquisition Analysis

In this section we analyze our learned acquisition schemes. Fig. 3 visualizes the per-frame non-Cartesian k-space undersampling schemes optimized across the training set for decay estimation. Each RF shot is parameterized by 513 learnable control points, and the full trajectory is obtained by spline interpolation between these points. As a result, local trajectory complexity (i.e. local curve variance) is controlled by how the learned control points

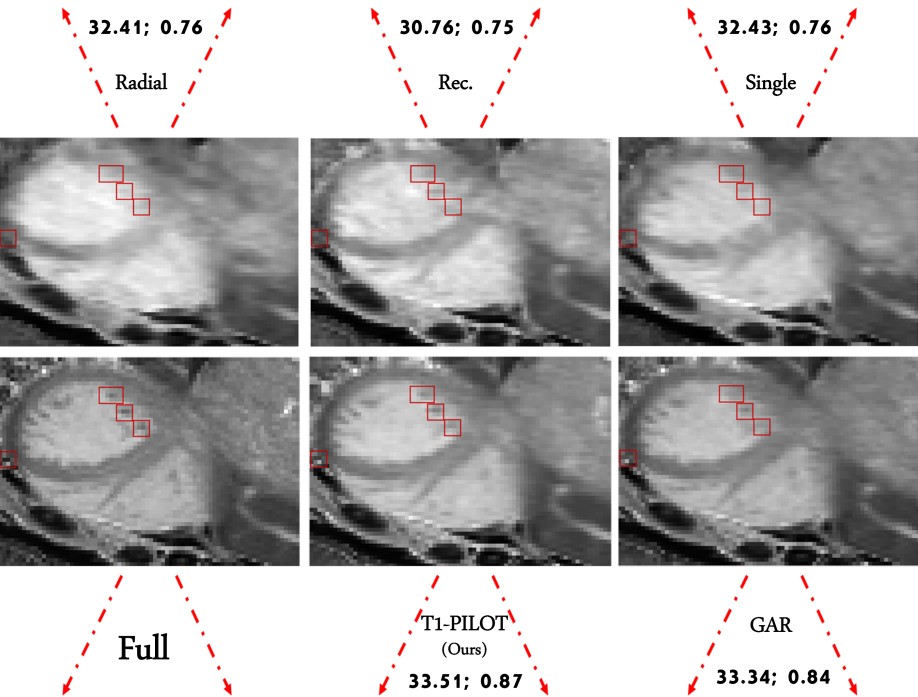

Figure 2: **Map Estimations across methods** - *Full* indicates map estimation without undersampling. For each baseline, we specify ROI PSNR (left) and VIF (right) compared to the fully-sampled map estimation. Our method's relative advantage over baselines is highlighted in red squares.

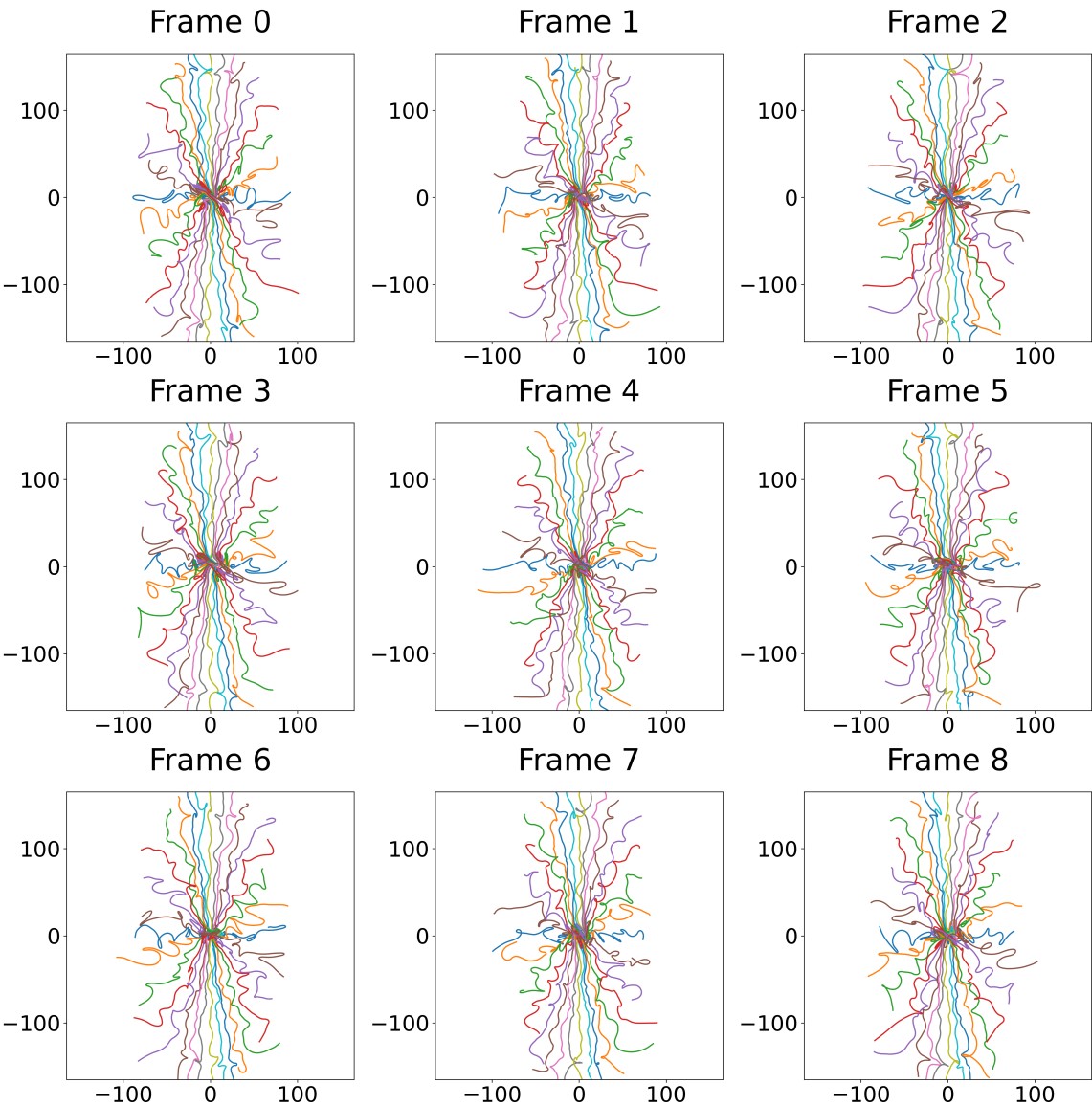

Figure 3: Learned non-Cartesian per-frame k-space trajectories using 16 shots per frame. Each colored curve corresponds to a single RF shot within a frame. Trajectories are parameterized by 513 learnable control points per shot and interpolated using splines.

are distributed: regions with rapidly varying control points allow high curvature and variability, while regions where control points change slowly yield smoother sampling paths.

A clear pattern emerges across all frames - trajectories exhibit higher variability near the k-space center and become progressively smoother as they move toward higher spatial frequencies. Since only the control points are learned and the rest of the curve is interpolated, this behavior indicates that the model deliberately concentrates its degrees of freedom around low-frequency regions. This reflects the fact that low spatial frequencies dominate the global signal evolution and are most informative for stabilizing T1 decay estimation. In contrast, high-frequency regions, which mainly encode fine spatial details, are sampled with smoother, more regular paths, indicating lower emphasis in the decay-driven optimization. When comparing the progression of learned sampling masks along time (early vs. later frames), we observe a second trend: in earlier frames, the central portions of the trajectories are more variable, while outer regions are smoother. As frames progress, the center becomes smoother and the outer regions exhibit increased variability. This suggests a temporal strategy learned by the model: early frames focus on accurately capturing the static, low-frequency structure of the scene, which is critical for establishing a stable baseline for decay fitting. Once this information is efficiently acquired, later frames allocate more modeling capacity to higher spatial frequencies, enabling better capture of subtle, dynamic details that refine the quantitative estimation.

Beyond these two main effects, the trajectories share a stable global geometry across frames, with consistent anisotropic structure and strong concentration near the k-space center. Variations across frames are structured rather than random, indicating that the model learns a coherent acquisition strategy that distributes complementary information over time rather than repeating identical patterns.

## 4. Conclusion

We presented T1-PILOT - a physically feasible accelerated T1 mapping algorithm. We showed that by explicitly integrating the T1 signal relaxation model into the undersampling–reconstruction pipeline, our approach learns multi-frame non-Cartesian k-space acquisition while harnessing data-driven priors to achieve both high acceleration and quantitative accuracy. Ablations confirm that incorporating the relaxation model outperforms fixed or single-frame schemes in comparison to the fully-sampled teacher reference, providing practical insights for designing T1 mapping strategies and broader quantitative MRI acceleration pipelines.

Despite these encouraging results, two limitations remain. First, the multi-stage optimization and added learnable parameters in our method introduce higher computational demands. Second, our work is focused on cardiac T1 mapping. Future work should expand our evaluation to broader modalities and datasets, and also study our method's performance under domain shifts such as different MR contrasts, scanner configurations, and acquisition centers. Motivated by our results and mentioned directions for future exploration, we believe T1-PILOT's framework can guide further research into rapid, model-aware imaging, ultimately helping to streamline quantitative MRI acquisition in clinical and scientific contexts.

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
