# OpenReview forum: "T1-PILOT: Physics-Informed Learned Optimized Trajectories for T1 Mapping Acceleration"
_MIDL.io/2026/Conference — MIDL 2026 Poster_

### Official Review · Reviewer_fG1w · 2026-01-07

**Confidence:** 5
**Preliminary Rating:** 4
**Final Rating:** 4

**Summary:**

This paper proposes T1-PILOT, a novel end-to-end framework for accelerated cardiac T1 mapping that explicitly leverages the physical T1 relaxation model during the joint optimization of k-space sampling trajectories and image reconstruction. After testing on public CMR dataset CMRxRecon, T1-PILOT outperforms other baseline methods.

**Strengths:**

1. Explicitly integrates the T1 relaxation model into optimization, leading to more reliable and physically consistent T1 estimation.
2. The approach outperforms multiple strong baselines across PSNR and VIF.

**Weaknesses:**

1. Cross-scanner and multi-center generalization were not explicitly evaluated.
2. The method assumes an accurate T1 relaxation model, while deviations from this model in real acquisitions may affect robustness.
3. Fully sampled k-space data needed.
3. Lack of the interpretability of learned trajectories.

**Detailed Comments:**

It would be beneficial if the learned sampling trajectories could be visualized for better interpretability.

**Justification Of Final Rating:**

Thanks for your effort. I am glad that the authors share such detailed explanation for me. You already answered all my questions. Now I would say I am happy to accept this paper. Looking forward for your final version. Thanks again!

**Justification Of The Preliminary Rating:**

My rating is based on the effective incorporation of the physical T1 relaxation model into the training framework. And the experimental validation on public dataset also shows the effectness although only based on single dataset.

**Questions To Address In The Rebuttal:**

No

---

> ### Author Response · Authors · 2026-01-18
>
> We thank reviewer fG1w for the helpful review of our work. We address the weaknesses raised.
>
> Regarding multi-scanner and multi-center generalization: we agree that cross-scanner and multi-center generalization, as well as other domain shifts, are important and are not evaluated in our current work. While CMRxRecon offers some variability (e.g., patients and protocols), it does not systematically capture strong distribution shifts across scanners or centers. We added this limitation to the discussion section.
>
> Regarding assumptions on the decay model: we agree that the exponential decay model is only an approximation and does not perfectly describe every real acquisition due to noise, motion, heart-rate variability, imperfect inversion, and other effects. Our method does not assume that the decay model is exact - rather, it is used as a structured regularizer and inductive bias to stabilize learning under noise, undersampling, and model mismatch.
>
> Regarding reliance on fully sampled data: like many learning-based MRI acceleration methods, our training uses fully sampled data to simulate undersampling. This limitation affects only training. At deployment, only the learned trajectories and trained networks are used, and no fully sampled data is required. We have revised the paper to describe our method as "retrospective simulation-based" rather than "self-supervised."
>
> Regarding trajectory interpretability: we thank the reviewer for this suggestion and have added visualizations and a dedicated discussion of learned trajectories in Section 3.5 of the revised paper.

---

### Official Review · Reviewer_4E53 · 2026-01-12

**Confidence:** 3
**Preliminary Rating:** 4
**Final Rating:** 4

**Summary:**

The paper introduces T1-PILOT, an end-to-end framework for accelerating cardiac T1 mapping by jointly optimizing k-space undersampling trajectories and image reconstruction while explicitly modeling the physical T1 signal relaxation process. The method employs a three-stage optimization strategy that progressively transitions from general image reconstruction to accurate T1 decay estimation. Extensive experiments on the CMRxRecon dataset demonstrate that T1-PILOT outperforms several baseline approaches, including learned single-mask methods as well as fixed radial and golden-angle sampling schemes.

**Strengths:**

Overall, this is a technically strong paper, and the explicit incorporation of the physical decay model is a noteworthy aspect of the approach. The authors compare their method against appropriate baseline strategies, and the inclusion of per-sample reconstruction and decay refinement further strengthens the pipeline by enabling improved patient-specific estimation.

**Weaknesses:**

A primary weakness of the paper is that, although it claims reduced acquisition times as a key advantage, this aspect is not explicitly demonstrated or quantitatively compared against other methods. Given the practical significance of acquisition time reduction, showcasing this benefit more clearly would strengthen the paper and better support the stated claims.

**Detailed Comments:**

Major Points:

(1) Please report the acquisition times for all baseline methods to enable a clearer comparison of practical efficiency.

(2) Please provide additional details on the number of trainable parameters for the learning-based baseline methods. While these models are expected to have higher parameter counts, explicitly reporting them would improve transparency and help readers better contextualize the comparisons.

Minor Points:

(1) Figure 1 is difficult to read in its current form, and additional explanation or improved clarity would be helpful.

**Justification Of Final Rating:**

I thank the authors for address all my concerns. I think this paper is technically strong paper, and the explicit incorporation of the physical decay model is unique and would be a useful methodology for researchers.

**Justification Of The Preliminary Rating:**

In its current form, the paper is acceptable; however, several edits and clarifications would improve the presentation and help readers better understand the methodology and its practical capabilities.

**Questions To Address In The Rebuttal:**

I would like all major and minor points to be addressed.

---

> ### Author Response · Authors · 2026-01-18
>
> We thank reviewer 4E53 for the careful analysis of our work.
>
> Regarding the lack of comparison to other T1-mapping baselines: we agree that such comparisons would be valuable, but many existing methods are not directly comparable to our setting. For example, [MyoMapNet](https://arxiv.org/abs/2104.00143) reduces scan time by using fewer predefined inversion-recovery images and learning to regress a T1 map from them, rather than accelerating each image via compressed sensing or learned k-space undersampling. Since it does not model k-space or optimize sampling trajectories, it is not directly comparable to our task of decay-aware acquisition design. Other related works such as [Paajanen et al.](https://www.mdpi.com/2313-433X/9/8/151) and [mcLARO](https://arxiv.org/abs/2304.03458) are complex to implement and do not provide reproducible public code. We therefore focused on baselines that are reproducible and isolate the effect of task-aware acquisition learning, and we hope our released code will help facilitate broader comparisons in future work.
>
> Regarding the request to report acquisition times: all methods compared at a given shot setting have identical nominal acquisition time, since they use the same number of frames and shots per frame. Differences in performance therefore reflect acquisition efficiency rather than longer scan duration. Our method achieves comparable or superior T1 accuracy at 16 shots compared to baselines at 32 or 64 shots, corresponding to approximately 2×–4× reduction in acquisition time for similar quantitative performance. We added a clarifying sentence in Section 3.3 of the revised version.
>
> Regarding the request to report the number of trainable parameters: a breakdown of trainable parameters for each learning-based baseline has been added to Section 3.2.
>
> Regarding the clarity of Figure 1: we added a more detailed caption to improve readability and guide the reader through the pipeline.

---

### Official Review · Reviewer_cHWX · 2026-01-13

**Confidence:** 3
**Preliminary Rating:** 4

**Summary:**

The authors propose an end-to-end framework that jointly learns non-Cartesian, per-frame k-space trajectories and reconstruction/decay estimation, but crucially optimizes the sampling trajectories using the downstream T1 decay objective, rather than using reconstruction loss alone.
Methodologically, they define a neural decay estimator that outputs decay parameters conditioned on the reconstructed sequence and inversion times, enforcing the exponential model during training (Eq. 1-3). Experiments on CMRxRecon compare against fixed radial, golden-angle ratio, a single learned mask shared across frames, and a reconstruction-only learned per-frame scheme.
They report improvements in PSNR/VIF and claim better preservation of fine myocardial structures (Table 1, Fig. 2).

**Strengths:**

There are a couple of strengths:

x. Right objective: task-aware sensing. The core ideas of learning acquisition trajectories using the decay-model fit as the training signal, sounds principled. It aligns measurement design with the quantitative endpoint (T1 fidelity), not merely intermediate image fidelity

x. Clear formulation with explicit physical inductive bias. The exponential relaxation model is enforced explicitly (Eq. 1–3), and the network outputs decay parameters rather than directly regressing T1 maps, which is a sensible bias for identifiability and regularization

x. Optimization schedule is great. Jointly learning (a) non-Cartesian trajectories and (b) a downstream nonlinear parameter fit is noisy and unstable; the proposed staged schedule (recon-first --> decay-aware) is a practical engineering contribution and well-motivated.

x. Evaluation includes both decay-consistency and map-consistency metrics. Reporting sequence-level decay fit (reconstructing) and map-level comparison gives two complementary views of performance (Table 1).

**Weaknesses:**

I also identify a few weaknesses.

x. No ground-truth T1 maps; “GT” is a model trained on fully sampled data. The paper states that because ground-truth T1 maps are unavailable, they train a decay-estimation model on fully-sampled data and treat its output as GT. It turns the evaluation partly into agreement with a teacher model, not accuracy to a physical or clinical reference. Improvements in “T1-Map PSNR/VIF” may reflect closer matching to that teacher’s bias.

x. Confounding from heavy pretraining and pipeline complexity. Reconstruction pretraining uses TEAM-PILOT pretrained on augmented OCMR, then further trained per experiment. **The method’s gains might partially hinge on this initialization**, but the paper does not clearly quantify “from scratch” performance or sensitivity.

x. Baselines are somewhat narrow for the claim. If the paper’s claim is “physics-informed trajectory optimization improves quantitative mapping,” the baseline set should include at least one strong quantitative mapping pipeline that does not rely on learned trajectory optimization but is competitive in map accuracy.

x. “Self-supervised” framing is partially convincing. The optimization uses fully sampled data to simulate undersampling during training and relies on reconstruction loss vs the fully sampled reference in stage (1). That is common in MRI ML papers, but it’s not “self-supervised” in the strict sense of learning from only acquired undersampled measurements with no fully-sampled reference. The terminology should be tightened.

**Detailed Comments:**

Please see my comments above.

Additionally:

x. Some phrasing overstates generality (“significantly outperforms… at greater acceleration factors”); the reported gains are meaningful but the evaluation target is not true ground truth. I would tone this down to “improves agreement with the fully-sampled-teacher reference and improves decay-consistency.”

**Justification Of The Preliminary Rating:**

The core idea seems to be the right direction, and the empirical gains are consistent (Table 1, Fig. 2).
But I would push the authors on evaluation validity (teacher-as-GT) and baselines. Without those clarifications, the paper risks feeling like a sophisticated pipeline that’s correct in spirit but not yet nailed down in evidence.

**Questions To Address In The Rebuttal:**

Please see my comments above.

---

> ### Author Response · Authors · 2026-01-18
>
> We thank reviewer cHWX for the thorough analysis and constructive critique of our work. We wish to address several of the weaknesses mentioned by the reviewer.
>
> Regarding the use of OCMR pretraining for TEAM-PILOT: the entire pretraining process took around 10 hours on a single NVIDIA RTX-2080 GPU. The majority of training time in our experiments is devoted to training on the target CMRxRecon dataset. Therefore, we do not consider this pretraining stage to be unusually heavy. Moreover, all baselines in our paper are initialized from the same OCMR-pretrained TEAM-PILOT checkpoint. This ensures that relative performance differences are strictly due to the decay and acquisition models being compared. In addition, OCMR pretraining does not provide a large performance boost over training directly on CMRxRecon -  its main benefits are faster convergence and reduced overfitting, which can otherwise harm both in-domain and out-of-domain performance.
>
> Regarding narrow baselines: we agree that this is a limitation of our paper. We kindly refer the reviewer to our response to reviewer 4E53, where we elaborate on the reasons for excluding additional comparisons.
>
> Regarding the framing of our method as "self-supervised": we accept this remark and thank the reviewer for improving the precision of our presentation. We have revised the paper to replace this term with "retrospective simulation-based."
>
> Regarding potentially overstated phrasing: we agree and have revised the manuscript to avoid overstatement. We now describe improvements as better agreement with the fully-sampled teacher reference as proposed.

---

> > ### Comment · Reviewer_cHWX · 2026-01-20
> > **thank you**
> >
> > Thank you for the response. The rebuttal has addressed most of my concerns.

---

### Author Rebuttal · Authors · 2026-01-18

**Rebuttal:**

T1-PILOT revision addressig reviewers' comments collectively.
Changes from initial submission are highlighted in red and are pointed in the official comments.

**Supporting Material:**

/attachment/20a58e8986c41cfe2edd4454fb17951eadf094b0.pdf

---

### Meta-Review · Area_Chair_fcgZ · 2026-02-04

**Recommendation:** Accept (Poster)
**Confidence:** 5

**Metareview:**

All reviewers found the proposed method to be novel and the results promising.

---

### Decision · Program_Chairs · 2026-02-13

Accept (Poster)